# The Effect of Synthetic Polyamine BPA-C8 on the Fertilization Process of Intact and Denuded Sea Urchin Eggs

**DOI:** 10.3390/cells13171477

**Published:** 2024-09-02

**Authors:** Nunzia Limatola, Jong Tai Chun, Jean-Louis Schmitt, Jean-Marie Lehn, Luigia Santella

**Affiliations:** 1Department of Research Infrastructures for Marine Biological Resources, Stazione Zoologica Anton Dohrn, 80121 Napoli, Italy; nunzia.limatola@szn.it; 2Department of Biology and Evolution of Marine Organisms, Stazione Zoologica Anton Dohrn, 80121 Napoli, Italy; chun@szn.it; 3Laboratory of Supramolecular Chemistry, Institut de Science et d’Ingénierie Supramoléculaires ISIS, Université de Strasbourg, 8 Allée Gaspard Monge, 67000 Strasbourg, France; jlschmitt@unistra.fr (J.-L.S.); lehn@unistra.fr (J.-M.L.)

**Keywords:** sea urchin eggs, extracellular matrix, polyamine, actin, calcium, fertilization, vitelline layer, sperm receptor, acrosome reaction

## Abstract

Sea urchin eggs are covered with layers of extracellular matrix, namely, the vitelline layer (VL) and jelly coat (JC). It has been shown that sea urchin eggs’ JC components serve as chemoattractants or ligands for the receptor on the fertilizing sperm to promote the acrosome reaction. Moreover, the egg’s VL provides receptors for conspecific sperm to bind, and, to date, at least two sperm receptors have been identified on the surface of sea urchin eggs. Interestingly, however, according to our previous work, denuded sea urchin eggs devoid of the JC and VL do not fail to become fertilized by sperm. Instead, they are bound and penetratedby multiple sperm, raising the possibility that an alternative pathway independent of the VL-residing sperm receptor may be at work. In this research, we studied the roles of the JC and VL using intact and denuded eggs and the synthetic polyamine BPA-C8. BPA-C8 is known to bind to the negatively charged macromolecular complexes in the cells, such as the JC, VL, and the plasma membrane of echinoderm eggs, as well as to the actin filaments in fibroblasts. Our results showed that, when added to seawater, BPA-C8 significantly repressed the Ca^2+^ wave in the intact *P. lividus* eggs at fertilization. In eggs deprived of the VL and JC, BPA-C8 binds to the plasma membrane and increases fibrous structures connecting microvilli, thereby allowing the denuded eggs to revert towards monospermy at fertilization. However, the reduced Ca^2+^ signal in denuded eggs was nullified compared to the intact eggs because removing the JC and VL already decreased the Ca^2+^ wave. BPA-C8 does not cross the VL and the cell membrane of unfertilized sea urchin eggs to diffuse into the cytoplasm at variance with the fibroblasts. Indeed, the jasplakinolide-induced polymerization of subplasmalemmal actin filaments was inhibited in the eggs microinjected with BPA-C8, but not in the ones bath-incubated with the same dose of BPA-C8.

## 1. Introduction

Gametes are specialized cells for sexual reproduction whose primary role is as free singular cells participating in fertilization rather than as stationary tissue constituents. For this reason, animal sperm have remarkable motility, and the eggs are endowed with enormous nutritional reservoirs and metabolic capability, as well as hard shells or envelopes to protect themselves. Oocytes and eggs are large, and a thick extracellular matrix with structural and physiological features surrounds their plasma membranes. As for sea urchin eggs, the outermost layer of the extracellular matrix is the jelly coat (JC), which is suggested to modulate the egg’s encounter with sperm by limiting or harboring the approaching sperm [1,2,3,4,5].

The JC of sea urchin eggs comprises a meshwork of fucose sulfate polysaccharides bound to sialoglycoproteins [6,7,8]. The JC also contains, and thereby releases, signaling peptides such as “speract” (*Strongylocentrotus purpuratus*) and “resact” (*Arbacia punctulata*) that are known to serve as chemoattractants for conspecific sperm [9,10]. The fucose sulfate polysaccharide of the egg JC is bound by its receptor on the sperm, which induces a Ca^2+^-dependent acrosomal reaction in the sperm in a species-selective manner owing to the identity of the polysaccharide, its sulfation pattern, and the position of its glycosidic linkage [7,11,12,13,14]. The acrosome reaction in sea urchin sperm is characterized by the exocytotic fusion of the acrosomal vesicle with the sperm plasma membrane, accompanied by the extension of a relatively short acrosomal process. As a result, the foremost tip of the sperm head becomes studded with the cell adhesion protein bindin that originates from the acrosomal vesicle [15,16,17,18]. The bindin-exposing sperm then adheres to the conspecific egg surface, partly owing to its selective binding to the receptor’s saccharide moiety residing in the egg’s vitelline layer (VL) [17,19,20,21].

Based on morphological and biochemical data, it has long been suggested that the sea urchin sperm binds to its conspecific egg at the VL [22,23,24,25,26,27,28], the fibrous extracellular matrix intimately associated with the egg plasma membrane [29,30]. Because a glycoprotein in the egg plasma membrane can extend its glycosylated portions to the VL, bindin is believed to bind to such a protein that serves as its receptor. However, some studies have raised the possibility that bindin can also directly bind to phospholipids in the plasma membrane [19,31]. According to a recent study using the sperm obtained from a transgenic sea urchin whose bindin gene was nullified by genome editing, bindin is essential for fertilization in sea urchins (*Hemicentrotus pulcherrimus*) [32]. On the other hand, the molecular mechanism by which the sperm binds to the sea urchin egg has been less precise regarding the identity of the receptor binding to bindin or another region of sperm. Earlier works demonstrated that bindin has a lectin-like activity, binding to polysaccharides [17], and at least two distinct receptors binding to sperm or bindin have been identified: a 350 kDa glycoprotein and EBR1 (egg bindin receptor 1) [33]. The 350 kDa protein is highly glycosylated (70%) and forms a homomeric polymer by intermolecular disulfide bonds, which is susceptible to reducing agents such as dithiothreitol (DTT) [34]. The immunoreactivity of the 350 kDa protein is enriched at the tip of microvilli on the surface of sea urchin eggs [35]. While the sperm interacts with both polypeptide and oligosaccharide chains of the 350 kDa receptor to promote species-selective gamete fusion [36,37], subtractive cDNA library screening to search for species-specific sea urchin mRNA led to the identification of EBR1, whose polypeptide region binds to bindin [38]. Although these two proteins satisfied the criteria as receptors for bindin or the sperm, whether or not these are the only sperm receptors on the sea urchin egg is an open question [33].

Interestingly, our previous work indicated that *Paracentrotus lividus* eggs whose JC and VL were removed by DTT and the alkaline treatment (“denuded eggs”) did not fail to become fertilized, but instead incorporated more than three sperm on average [39]. If a sperm receptor equivalent to the 350 kDa glycoprotein or EBR1 is the primary sperm interaction point in *P. lividus* eggs, the disruption of its disulfide bonds amid the removal of the VL altogether should have inhibited the fertilization process, as was reported with *S. purpuratus* [34]. After removing the JC and the VL, the denuded egg of *P. lividus* still retained the barrier against heterospecific sperm (*Arbacia lixula*), but were rendered more permissive to fuse with multiple sperm of its species [39]. These observations suggest that there exists an alternative mode of sperm entry independent of the two identified sperm receptors in the VL, and that a significant function of the VL may be to limit sperm fusion, which challenges the conventional view on sperm egg interaction. Hence, there is more to learn about the roles of the VL as a part of the molecular mechanisms of species-specific gamete interaction and sperm entry.

Whereas the VL intimately covers the egg, the inner face of the plasma membrane is closely associated with the prominent meshwork of the actin cytoskeleton whose structural remodeling helps to change the shape of the cell periphery, such as microvilli [40,41,42,43]. In the tight junctions flanking the plasma membrane, the extracellular events on the VL may be relayed to the actin cytoskeleton and various signaling components of the cytoplasm by way of proteins like integrin [44,45,46]. The sperm-induced structural and functional changes in the actin filaments within the microvilli of the egg surface and cortex are interesting. Indeed, it has been shown that the fine regulation of the actin cytoskeleton of the egg cortex is an essential prerequisite for the successful monospermic fertilization response, as repeatedly demonstrated in many animal species, including clams, starfish, and sea urchins [47,48,49,50,51,52]. In starfish and sea urchins, the control of structural actin dynamics at the unfertilized egg cortex is significant. Its deregulation has profound effects on sperm-induced Ca^2+^ signaling, often leading to polyspermy or failure of sperm entry into the egg [49,53]. At the site of sperm binding, the egg’s actin filaments within the microvilli and the subjacent cortical region reorganize themselves to form the fertilization cone, achieved through polymerization of the actin meshwork intimately associated with the egg plasma membrane [53,54,55,56,57]. The depolymerization and subsequent re-polymerization of the subplasmalemmal actin filaments are instrumental in facilitating cortical granule exocytosis and the extension of microvilli in the perivitelline space of the sea urchin and starfish eggs during fertilization [58,59,60]. This research underscores the critical role of both the extracellular matrix (JC and VL) and the subplasmalemmal actin filaments in regular fertilization and sperm entry into sea urchin eggs, providing valuable insights into the intricate process of fertilization.

Recently, we demonstrated that a branched polyamine, N,N,N’,N’-[tetrakis(aminopropyl)-octamethylenediamine (herein referred to as BPA-C8), can serve as an effective marker for the JC, VL, and plasma membrane in sea urchin eggs and starfish oocytes [55,56] because the amine groups of the polyamine are protonated at physiological pH, thereby binding anionic complexes such as the sulfated polysaccharides in the extracellular matrix. The polycationic nature of natural and synthetic polyamine also accelerates the polymerization of the actin molecules. It may even promote crosslinking between the actin filaments *in vitro* [61,62,63]. Moreover, it was shown that BPA-C8 added to fibroblasts in culture stimulated lamellipodia formation by stabilizing actin filaments [64]. BPA-C8 also interferes with the TGF-β1-mediated conversion of fibroblast into myofibroblast [65]. On the other hand, it was demonstrated that 1–5 mM of natural polyamines, i.e., spermine and spermidine, can induce the polymerization of purified fibronectin into filamentous structures in the condition of low ionic strength [66]. As fibronectin is a high-molecular-weight glycoprotein, these observations together suggest that both natural and synthetic polyamines, in general, tend to favor the formation or stabilization of anionic macromolecular complexes such as extracellular matrix, plasma membrane, actin filaments, and nucleic acids such as DNA and RNA [61,62,63,67,68,69,70,71,72].

To explore the utility of synthetic polyamine and to study the biological function of the JC and VL upon fertilization of the sea urchin eggs, we thoroughly examined the effect of BPA-C8 on intact and denuded sea urchin (*P. lividus*) eggs. Our previous studies suggested that BPA-C8 might not traverse the plasma membrane of the unfertilized eggs to diffuse into the starfish and sea urchin eggs, as indicated by the subcellular distribution of the fluorescently labeled BPA-C8 [55,56]. To ensure a robust comparison, BPA-C8 (not fluorescent) was administered to the eggs by bath incubation and microinjection. Given that BPA-C8, as a polyamine, is expected to influence both intracellular and extracellular structures, as well as Ca^2+^ channels [73,74,75,76], we meticulously analyzed the structure of these BPA-C8-treated eggs (by confocal and electron microscopy), Ca^2+^ signals, and the morphology of the egg cortical actin cytoskeleton before and after fertilization, ensuring the validity and reliability of our findings.

## 2. Materials and Methods

### 2.1. Gamete Preparation and Fertilization

Sea urchins *Paracentrotus lividus* and *Abarcia lixula* were collected in the Gulf of Naples from December to May and from May to September, respectively, and maintained in a tank with circulating seawater at 16 °C. Spawning was induced by intracoelomic injection of 0.5 M KCl. Individual eggs were collected in natural seawater (NSW) filtered with a Millipore membrane (pore size 0.2 μm, Nalgene vacuum filtration system, Thermo Fisher Scientific, Rochester, NY, USA). To obtain denuded eggs, *P. lividus eggs* were incubated for 20 min in NSW containing 10 mM DTT with the pH adjusted to 9.0 using NaOH [39]. For fertilization, dry sperm were collected from the male animal and kept at 4 °C until use. Sperm were diluted in NSW only a few minutes before fertilization at the final concentration of 1.84 × 10^6^ cells/mL.

### 2.2. Visualization of the Egg-Incorporated Sperm

*P. lividus* sperm diluted in NSW were stained with 5 µM Hoechst-33342 (Sigma-Aldrich, St. Louis, MO, USA) for 30 s before fertilization. The number of sperm incorporated into fertilized eggs was counted 5 min after insemination by epifluorescence microscopy with a cooled CCD (charge-coupled device) camera (MicroMax, Princeton Instruments, Inc., Trenton, NJ, USA) mounted on a Zeiss Axiovert 200 inverted microscope (Carl Zeiss AG, Oberkochen, Germany) with a Plan-Neofluar 40×/0.75 objective and a UV laser. The Hoechst-33342 solution used in this condition could visualize the decondensed male pronucleus in the zygote.

### 2.3. Scanning Electron Microscopy (SEM)

Untreated and BPA-C8-treated unfertilized *P. lividus* eggs and zygotes were fixed in NSW containing 0.5% glutaraldehyde (pH 8.1) for 1 h at room temperature and post-fixed with 1% osmium tetroxide for another hour. The specimens were dehydrated in ethanol with increasing concentrations and were subjected to critical point drying (LEICA EM CP300). The samples were coated in a thin layer of gold using a LEICA ACE200 sputter coater and were observed with a JEOL 6700F scanning electron microscope (Akishima, Tokyo, Japan).

### 2.4. Transmission Electron Microscopy (TEM)

For TEM observations, the samples were fixed directly in NSW containing 0.5% glutaraldehyde (pH 8.1) for 1 h at room temperature and were post-fixed with 1% osmium tetroxide and 0.8% K_3_Fe(CN)_6_ for another hour at 4 °C. After washing two times in NSW, the samples were rinsed twice in distilled water for 10 min and subsequently treated with 0.15% tannic acid for 1 min at ambient temperature. After extensive rinsing in distilled water (10 min, three times), the specimens were dehydrated in increasing acetone concentrations. They transferred to a mixed acetone solution and SPURR before being embedded in SPURR. Resin blocks were sectioned with a Leica EM UC7 ultramicrotome, and the ultrathin sections (70 nm in thickness) were observed without staining with a transmission electron microscope (Zeiss LEO 912 AB).

### 2.5. Chemicals, Reagents, and Recombinant Proteins

Jasplakinolide (JAS) was purchased from Invitrogen (Thermo-Fisher) and dissolved in DMSO. Polyamine BPA-C8 and its fluorescent derivative BPA-C8-Cy5 were synthesized following previously specified procedures [55,56,64]. Unless specified otherwise, Hoechst-33342 and all other unspecified materials in this study were purchased from Sigma Aldrich. The recombinant protein LifeAct-GFP was bacterially expressed [39] from the plasmid kindly provided by Dr. A. McDougall of Sorbonne University, France.

### 2.6. Microinjection, Ca^2+^ Imaging, and Confocal Microscopy

Intact unfertilized eggs were microinjected with an air pressure transjector (Eppendorf FemtoJet, Hamburg, Germany), as previously described [55]. For Ca^2+^ imaging, 500 µM Calcium Green 488 conjugated with 10 kDa dextran was mixed with 35 µM Rhodamine Red (Molecular Probes, Eugene, OR, USA) in the injection buffer (10 mM Hepes, 0.1 M potassium aspartate, pH 7.0) and microinjected into the eggs before insemination. The fluorescence signals of the cytosolic Ca^2+^ were acquired with a cooled CCD camera (Micro-Max, Princeton Instruments) mounted on a Zeiss Axiovert 200 microscope with a Plan-Neofluar 40×/0.75 objective at about 3 s intervals. The data were analyzed with MetaMorph (Universal Imaging Corporation, Molecular Devices, LLC, San Jose, CA, USA). Following the formula F_rel_ = [F − F_0_]/F_0_, where F represents the average fluorescence level of the entire egg and F_0_ is the baseline fluorescence, the overall Ca^2+^ signals were quantified for each moment. F_rel_ was expressed as RFU (relative fluorescence unit) for plotting the Ca^2+^ trajectories.

### 2.7. Visualization of Actin Filaments

LifeAct-GFP (10 µg/µL, pipette concentration) was microinjected into living eggs to visualize F-actin before and after fertilization at different experimental conditions. To visualize the plasma membrane, unfertilized eggs were incubated with 5 µM FM 1-43 (ThermoFisher Scientific). The eggs treated with the fluorescent probes were observed with a Leica TCS SP8X confocal laser scanning microscope equipped with a white light laser and hybrid detectors (Leica Microsystem, Wetzlar, Germany). The images were analyzed with MetaMorph to quantify the intensity of the LifeAct-GFP signals at the subplasmalemmal zones. To this end, fluorescence was line-scanned through the midline of the confocal plane, and the sum of the fluorescence in the region of interest, which was defined as 6 µm around the peak, was compared for the same egg before and after the BPA-C8 or JAS treatment.

### 2.8. Statistical Analysis

The numerical MetaMorph data were compiled and analyzed with Excel (Microsoft Office 2010) and reported as mean ± standard deviation in all cases in this manuscript. A one-way ANOVA and a U-test were performed through Prism 5.0 (GraphPad Software), and *p* < 0.05 was considered to be statistically significant. For ANOVA results showing *p* < 0.05, the statistical significance of the difference between the two groups was assessed by Tukey’s post hoc tests. For egg-incorporated sperm count analysis, the Mann–Whitney U test was carried out (https://www.socscistatistics.com/ accessed on 11 March 2024), and the two groups of data showing significant differences were marked with brackets and symbols indicating the *p*-values. The pairwise comparison that produced insignificant *p*-values (>0.05) was not mentioned for the sake of simplicity in the description.

## 3. Results

### 3.1. Effect of BPA-C8 on the Surface of Unfertilized and Fertilized Sea Urchin Eggs Visualized with Electron Microscopy

Our experimental process, utilizing both transmission and scanning electron microscopy, revealed the effect of BPA-C8 on the egg surface in altering the ultrastructure of the extracellular matrix and the egg cortex. The transmission electron microscopy (TEM) micrograph of unfertilized eggs without any treatment exhibited regularly spaced microvilli (MV), which are intimately associated with the vitelline layer (VL) (Figure 1A).

Another characteristic feature of the unfertilized egg is the positioning of cortical granules (CGs) at the subplasmalemmal region. When these eggs are fertilized, the CGs undergo exocytosis, and their inner contents are released into the perivitelline space (PS). This process promotes the separation of the VL from the egg plasma membrane, forming a fertilization envelope (FE) (Figure 1A, right panel). The subsequent cortical actin cytoskeleton polymerization phase included MV extension into the expanding perivitelline space.

On the other hand, in the eggs preincubated with BPA-C8 for 20 min, the extracellular matrix underwent structural alteration in a dose-dependent manner. Whereas the jelly coat (JC) of a sea urchin egg is usually lost during fixation, the pretreatment of the eggs with BPA-C8 preserved some of the JC in the vicinity of the VL. Albeit faint, the remnant of JC begins to appear in the eggs pretreated with 100 µM BPA-C8 (Figure 1B, unfertilized, arrowheads). Still, it was not appreciable over the elevating FE (Figure 1B, fertilized). In the eggs pretreated with 1 mM BPA-C8, the layer of dark dots covered the VL as if an extracellular layer as thick as 380 nm, in which the individual dots measured about 55 nm (Figure 1C, unfertilized, arrowhead). Upon sperm addition (5 min after insemination), this thicker outmost layer was often detached from the elevating FE (Figure 1C, fertilized, arrowhead). The new egg surface layer produced in the eggs pretreated with a high dose of BPA-C8 does not become a part of the FE. It may, thus, represent the remnant of the JC that was retained by the action of BPA-C8 as a molecular linker (discussed later). At the same time, the exogenous polyamine did not precipitate in seawater.

What was observed with TEM was also corroborated with scanning electron microscopy (SEM) (Figure 2).

Again, fixation of intact unfertilized eggs tended to remove the JC, so the SEM image displays the egg surface studded with MV spaced regularly (Figure 2A, unfertilized). After fertilization (5 min), these MV were no longer visible because of the elevated FE that masked the cell surface (Figure 2A, fertilized). Interestingly, the surface of the eggs pretreated with 100 µM BPA-C8 showed subtle changes around MV in the sense that some fibrous structures bridged the neighboring individual MV (Figure 2B, unfertilized), which was similar to the fibrous strands that connect microvilli of other species of sea urchin eggs [40]. Nonetheless, the elevation of FE appeared normal, and, consequently, the MV were again veiled by the elevated FE (Figure 2B, fertilized). In the eggs pretreated with 1 mM BPA-C8, the MV on the egg surface were entirely covered by the fibrous meshwork (Figure 2C, unfertilized). Therefore, the electron-dense outmost layer in the TEM image of the egg pretreated with 1 mM BPA-C8 shown in Figure 1C (arrowhead) is likely to correspond to the fibrous strands connecting the microvilli in the SEM image of Figure 2. When fertilized, some sperm were often seen to adhere to the egg surface (Figure 2C, fertilized). This is because the surface of these eggs is covered not only by FE, as shown in Figure 1C, but also by the condensed JC, which is thought to be sticky.

It has been known that the JC of sea urchin eggs is difficult to visualize due to its solubility in seawater. The color stain on the JC is usually short-lived, and the existence of the JC was only indirectly demonstrated by the exclusion of dyes like Indian ink in the media [77,78]. Whereas the JC of starfish is retained during the fixation for EM, the JC of sea urchins is always lost. As aforementioned, the remnant of JC visible in the TEM and SEM of intact sea urchin eggs pretreated with BPA-C8 (Figure 1 and Figure 2) might signify the retention of the JC that would have been lost by dispersion in seawater and fixatives. This preservation may arise from the fact that BPA-C8, as an organic polycation with a short carbohydrate chain, is likely to strengthen the structure of the JC by bridging the polysaccharide fibers of the JC adhering to the VL. In favor of the latter interpretation, when the eggs of another sea urchin species, *Arbacia lixula*, were incubated with the BPA-C8 but tagged with a fluorescent dye, i.e., BPA-C8-Cy5 (25 µM), the fluorescent BPA-C8 not only bound and visualized the JC, but also prevented the arrival of the sperm to the egg surface (Appendix A, unfertilized and fertilized, n = 12). However, when the BPA-C8-Cy5 concentration was lowered to 10 µM, the sperm traversed the JC. They reached the egg surface, as judged by the heavy staining on the rim of the egg (Appendix A, n = 11), representing FE, which is moderately elevated in this species [79]. The presence of the JC in the *A. lixula* egg not visible under light microscopy was also indirectly demonstrated by the eggs fertilized in the presence of the membrane-specific dye FM 1-43. In these eggs, owing to the enmeshing nature of the JC, multiple sperm are visible in the entire span of the JC (Figure 1C, SP, n = 17) and eventually succeed in fertilizing the egg.

### 3.2. Effect of BPA-C8 on the Denuded Eggs of P. lividus Visualized with Electron Microscopy

The JC and VL of sea urchin eggs can be chemically removed by incubating the eggs in the presence of 10 mM DTT at pH 9 [59,80]. These denuded eggs now lack the extracellular matrix that shields the eggs. To test the direct effect of BPA-C8 on the cell surface structure below the plasma membrane (PM), we treated the denuded eggs with BPA-C8 by bath incubation. The analysis of the ultrastructure by TEM showed that the morphology and the number of microvilli (MV) on the surface of the denuded eggs are appreciably changed due to the disruption and removal of the JC and VL [59]. The MV appeared sparser (Figure 3A, unfertilized) than those in the intact eggs (Figure 1A). Five minutes after fertilization, the elongated MV emanated from the cell surface. Nevertheless, the fertilization envelope (FE) failed to separate from the egg surface because of the lack of its precursor, the VL (Figure 3A, fertilized).

When the denuded eggs were preincubated with 100 µM BPA-C8 for 20 min, the tips of individual MV appeared bulged (Figure 3B, unfertilized). When fertilized, the MV of these treated eggs extended prominently longer than in the denuded eggs fertilized without prior exposure to BPA-C8 (Figure 3B, fertilized). The TEM image of the unfertilized denuded eggs preincubated with 1 mM BPA-C8 showed that the surface of the plasma membrane and the sectioned MV were intermittently studded with the electron-dense structure of about 30 nm in size (Figure 3C, unfertilized). At fertilization, it was frequently observed that some cortical granules (CGs) are nearly bound to the PM, but have difficulties completing exocytosis (Figure 3C, fertilized). Since both MV extension and CGs exocytosis depend on the fine regulation of the actin cytoskeleton in the egg surface, these results suggest that both egg denudation and BPA-C8 exposure alter the morphology and function of the actin filaments at the egg surface.

The denuded eggs, treated with BPA-C8 under the same experimental conditions, were subjected to SEM to validate our hypothesis (Figure 4).

As expected from the TEM image (Figure 3A), the denuded eggs exhibited a more scattered distribution of MV on the egg surface (Figure 4A, unfertilized) compared to those in intact eggs (Figure 2A). At fertilization, these MV extended and curved on the egg surface (Figure 4A, fertilized), lacking a FE. When the denuded eggs were incubated with 100 µM BPA-C8 for 20 min, the tip of individual MV appeared significantly swollen (Figure 4B, unfertilized), confirming the observation with TEM (Figure 4A, unfertilized). The fibrous bridges connecting the neighboring MV were also visible in SEM, although less pronounced (Figure 4B, unfertilized).

At fertilization, the denuded eggs pretreated with 100 µM BPA-C8 displayed a significantly altered pattern of MV extension (Figure 4B, fertilized). The tips of the MV were often flattened, indicating that the MV tips were generally swollen before fertilization. Similar results were obtained with the denuded eggs exposed to 1 mM BPA-C8 (Figure 4C, unfertilized), but the egg-adhering sperm were more frequently encountered after fixation (Figure 4C, fertilized).

### 3.3. Structure of the Actin Cytoskeleton in Sea Urchin Eggs Treated with BPA-C8

The altered morphology of MV, which are filled with actin filaments, in denuded eggs and the eggs treated with BPA-C8 prompted us to examine the changes of the actin filaments by use of LifeAct-GFP, a fluorescent probe visualizing filamentous actin (F-actin) before and after fertilization [39,59,81]. When viewed 20 min after microinjection into living and intact unfertilized *P. lividus* eggs, LifeAct-GFP visualized F-actin predominantly in the subplasmalemmal zone (Figure 5A, upper row, n = 11).

In the presence of 100 µM BPA-C8, the distribution of the F-actin visualized by LifeAct-GFP was not appreciably changed near the plasma membrane, but the signal was considerably reduced when the eggs were incubated in the presence of 1 mM BPA-C8 (Figure 5A, upper row, right panel, n = 10). On the other hand, the F-actin visualized in the denuded eggs of the same batch showed an appreciable reduction in its LifeAct-GFP signal intensity (Figure 5A, lower panels, n = 8). When BPA-C8 was added to the denuded eggs, the LifeAct-GFP signals at the subplasmalemmal zone appeared intermitted regardless of the BPA-C8 doses (Figure 5A, lower row, n = 6 and n = 7).

When the intact eggs were fertilized, the subplasmalemmal actin cytoskeleton underwent drastic reorganization [39]. By 5 min after insemination, the LifeAct-GFP signal became more intense near the plasma membrane (Figure 5B, upper low, left, n = 11). While the cortical actin cytoskeleton polymerization concomitant with the fertilization cone formation (arrow) was detected 5 min after insemination in eggs that had been incubated with 100 µM BPA-C8 for 20 min (n = 7), a thinner F-actin cortical layer was instead visible in those treated with 1 mM BPA-C8 and fertilized (n = 10). Occasionally, when the eggs were inseminated after 20 min preincubation with 1 mM BPA-C8, the FE did not elevate in an equidistance mode as in the control (Figure 5B, upper row). On the other hand, the denuded eggs displayed an enhanced presence of actin filaments in the subplasmalemmal zone, but the signal intensity was appreciably reduced in comparison with the intact egg (Figure 5B, lower row).

Interestingly, the denuded eggs tended to be polyspermic at fertilization [39], and multiple fertilization cones happened to be captured on the same confocal plane (Figure 5B, lower row, left panel, arrows). Since the VL was removed, the denuded eggs showed no formation of FE, unlike the intact eggs (Figure 5B, upper row). These results show that JC and VL removal significantly reduces the actin filaments near the plasma membrane. A high dose of BPA-C8 (1 mM) added to the seawater diminishes the subplasmalemmal actin filaments.

### 3.4. Effect of BPA-C8 on Actin Dynamics in Sea Urchin Eggs

The intracellular actin pool in sea urchin eggs is readily shifted to polymerization when the eggs are incubated with the actin drug jasplakinolide (JAS) [49]. Whether a cell responds to JAS would be a criterion for the functionality of its actin cytoskeleton. Thus, to examine the effect of BPA-C8 on this type of JAS-induced actin dynamics, sea urchin eggs microinjected with LifeAct-GFP were treated with BPA-C8 either by seawater incubation or by microinjection and tested for their capability to respond to JAS. As aforementioned, the F-actin signals in the subplasmalemmal zone were reduced after the incubation with 1 mM BPA-C8 (Figure 5), yet their response to JAS was still close to a two-fold increase, which was similar to the response displayed by other eggs exposed to 0 μM or 100 μM BPA-C8 (Figure 6A, Table 1, n = 6 and n = 12). To probe more directly the effect of the polyamine in the cytoplasm, the BPA-C8 was microinjected into the eggs to make equimolar concentration in the cytosol (100 µM and 1 mM) (Figure 6B). At 100 µM BPA (n = 6), the distribution of F-actin was not much different from that of the intact egg that did not receive BPA-C8 (n = 7). The localization of F-actin is predominantly at the subplasmalemmal region, and the LifeAct-GFP signal intensified in response to 12 µM jasplakinolide (JAS), which hyperpolymerized subplasmalemmal actin at the expense of the cytoplasmic actin (Figure 6A,B).

On the other hand, at a higher dose of BPA-C8 (1 mM), the LifeAct-GFP signal was more significantly reduced near the plasma membrane, which was barely above the level of the inner cytoplasm. The intensity of the LifeAct-GFP fluorescence signals at the subplasmalemmal region (i.e., the extent of actin polymerization) showed nearly no change despite the 15 min incubation with JAS (Figure 6B and Table 1, n = 10). Thus, it appears that, at higher doses, BPA-C8 affects the actin pool of the egg by interfering with the balance between the filamentous and globular actin.

### 3.5. Effect of BPA-C8 on Sperm Entry into the Intact and Denuded Eggs

To study the effect of BPA-C8 on polyspermic fertilization experienced by denuded eggs [39], the sea urchin eggs were inseminated with fresh sperm stained with Hoechst-33342 after dilution in NSW (Figure 7). The number of egg-incorporated sperm was counted using epifluorescence microscopy (Figure 7A). As for the intact eggs, BPA-C8 preincubation before insemination did not make much difference at 100 µM and 1 mM, regardless of the method of the polyamine treatment, i.e., bath incubation or microinjection. Indeed, the eggs examined 5 min after insemination turned out to be predominantly monospermic, as only one sperm entered the egg in most cases (Figure 7, arrowheads and Table 2). However, the insemination of the denuded eggs with the same concentration of sperm consistently led to polyspermy, as judged by the visualization of the fluorescent DNA of many sperm in their cytoplasm, confirming our previous report that the removal of the VL renders *P. lividus* eggs polyspermic at fertilization [39]. On average, nearly eight sperm entered the denuded eggs (Figure 7B and Table 2), implying that the egg lost control against polyspermy.

Surprisingly, when 100 µM BPA-C8 was added to the denuded eggs (20 min incubation) before insemination, most of them were penetrated by only one sperm. A higher effect was observed with the denuded eggs pretreated with 1 mM BPA-C8, where sperm entry was inhibited in 12 out of 40 (Table 2). Thus, the sperm receptivity of the denuded egg surface was reverted towards that of the intact eggs. When BPA-C8 was delivered by microinjection into intact eggs, which were subsequently deprived of the VL, the strong tendency of polyspermy displayed by denuded eggs (11 ± 1.35 sperm per egg) was significantly mitigated to 3.7 ± 1.55 sperm per egg for 100 μM BPA-C8 (and 5.85 ± 1.33 for 1 mM BPA-C8), but not due to monospermy (Table 2). These findings suggest that the effect of BPA-C8 on denuded eggs alleviating polyspermy is ascribable to what the polyamine does on the actin dynamics of the altered egg surface of denuded eggs (Figure 4A) rather than its action inside the egg.

### 3.6. Effect of BPA-C8 on Ca^2+^ Signals in Intact and Denuded Eggs at Fertilization

Given that polyamine influences the activities of the Ca^2+^ ion channel [74,75], it was interesting to investigate whether exogenous polyamine BPA-C8 can alter the Ca^2+^ responses displayed by sea urchin eggs during the fertilization process. At fertilization, sea urchin eggs manifest immediate and synchronous Ca^2+^ increase in the cortex (referred to as “cortical flash” (CF)) and the progressive Ca^2+^ wave (CW) that sweeps through the cortex from the sperm interaction site to the antipode of the egg. As shown in Figure 8A and Table 3, when intact eggs of *P. lividus* were pretreated with BPA-C8 by bath incubation, the peak amplitude of the CW was significantly reduced by BPA-C8. In contrast, the average peak amplitude of the CF resulting from Ca^2+^ influx was not affected. When the same experiment was performed with denuded eggs, the CF and the CW were significantly reduced, together with the morphological modification of microvilli, as was observed in the egg in which the VL was removed [39]. The suppressive effect of BPA-C8 on the CW was abolished, while the amplitude of CF was significantly reduced (Figure 8B, Table 3). The differential impact of extracellular polyamine BPA-C8 on the CF may be derived from the structural organization of microvilli, which is responsible for the pattern of the cortical Ca^2+^ release (CF) [82]. Microvilli covered by the VL in intact eggs could be “protected” by the action of polyamine, as compared to the altered microvillar morphology in denuded eggs (Figure 2 and Figure 4). On the other hand, the eggs microinjected with 1 mM BPA-C8 displayed a significant reduction in the amplitudes of both CF and CW, and the eggs microinjected with a lower dose (100 μM) only exhibited a milder effect on CW (Figure 8C, Table 3). Hence, it appears that the activity of different kinds of Ca^2+^ channels localized on microvilli and, intracellularly, the CF and CW is commonly affected by a high dose of BPA-C8 due to the dramatic alteration of the egg actin cytoskeleton dynamics (Figure 6).

### 3.7. Sperm Entry during the Fertilization Envelope Elevation at the Early Fertilization Stage in Sea Urchin Eggs

Our previous findings that the intact structure of the VL of the unfertilized egg is essential to prevent polyspermy [39] prompted an investigation to survey the early ultrastructural and topographic changes of the egg surface at the sperm entry site. Intact sea urchin eggs were quickly subjected to fixation about 10 s after their insemination in NSW, a period during which the fertilizing sperm succeeds in fusing with the egg at the time of the initiation of the separation of the VL from the egg plasma membrane. Because the extent of the fertilization process progressing in the individual eggs before being arrested by the fixatives varied slightly in time, it was possible, for the first time, to compile a sequence of events featuring sperm entry in several fertilized eggs (Figure 9). The TEM image of a fertilized egg of *Arbacia lixula* showed a sperm being ushered into the egg by the fertilization cone (Figure 9A), which is enriched with actin filaments [53,55,56,83] (Figure 5). It is interesting to note that since the elevation of FE in the *A. lixula* egg is not as high as in *P. lividus* [79], it was possible to capture the moment of the passage of the fertilizing sperm through the lifted VL from the egg plasma membrane forming the perivitelline space (PS). The SEM micrographs reveal a significant stage of sperm entry in various *P. lividus* fertilized eggs, demonstrating a higher VL separation under natural physiological conditions, a departure from the previous practice of affixing them to polylysine-coated dishes [84,85]. Note that the fibrous material in the PS (arrow) deriving from the CGs exocytosis (shown in D and E, arrowheads) is visible behind the flipped FE (asterisk) that was ruptured during the fixation (Figure 9B). Notably, the hole (asterisks) on the partially elevated FE through which the sperm passes to enter the egg is apparently “elastic” enough to let in its large head (Figure 9B,C) and small enough to make a tight closure around the piercing tail of the sperm (Figure 9B–F, asterisk). It is also noteworthy that the engulfed sperm is fused with the activated egg, undergoing cortical depression or flattening [55,58,86,87] as judged by its presence in the PS behind the FE that was flipped open during the fixation (Figure 9E,F, asterisk).

## 4. Discussion

Naturally occurring polyamines are organic polycations widely distributed in animals, but their precise roles are much less understood than other organic molecules such as amino acids or nucleic acids. Putrescine, spermidine, and spermine are the most common polyamines enriched in mammalian brains [69,88,89,90]. Spermine is also abundant in sea urchin eggs, and the spermidine level sharply increases during cleavage cycles and gastrulation. These polyamines are presumed to play some roles during embryonic development, but detailed mechanistic information is missing [91,92,93,94]. When considering polyamine as a molecular linker bridging anionic portions of a variety of macromolecules, it is no surprise that its homeostatic deregulation is implicated in many biological conditions such as the impairment of actin-dependent cell motility and growth, apoptosis, and neurodegenerative diseases like Parkinson’s disease [68,95]. The enzyme metabolizing polyamine is also a strategic target for slowing cancer growth [96].

In this study, using *P. lividus* eggs, we studied the exogenous effect of synthetic polyamine BPA-C8, which has been utilized in cell biological research on fibroblasts [64,65]. First of all, we noted that, at variance with what was observed in cultured fibroblasts [64], BPA-C8 did not appear to penetrate the thick cell membranes of sea urchin eggs. Previously, we reported that fluorescently tagged BPA-C8 added to the incubation media and strongly labeled the JC, VL, and the plasma membrane of starfish oocytes and sea urchin eggs. By contrast, the fluorescence signal was not observed in the cytoplasm of these oocytes and eggs except for the scattered internal membrane structures likely to represent endocytosed vesicles originating from the plasma membrane [55,56]. Hence, owing to the thick layers of extracellular matrix enriched with polysaccharide, sea urchin eggs are not permeant to BPA-C8, as was corroborated by BPA-C8-Cy5 (Appendix A). In large part, this may be attributed to the fact that the thick layer of negatively charged sulfated carbohydrates in the JC, VL, and plasma membrane traps the polyamine, implying that the extracellular matrix layers of the unfertilized egg may serve as a chemical barrier protecting the sea urchin egg, which, however, does not prevent the binding and fusion of the fertilizing sperm with the egg plasma membrane ([39] and this contribution). Supporting the idea of membrane impermeability, the eggs incubated or microinjected with BPA-C8 did not exhibit the same cytoskeletal effects, although the same doses of BPA-C8 were administered (Figure 6A,B). Although the treatments with 1 mM BPA-C8 visibly inhibited the JAS-mediated actin polymerization kinetics, the molecular mechanism by which BPA-C8 interferes with actin dynamics inside the sea urchin egg is not known. Based on the effect of natural polyamine (spermine and spermidine) and BPA-C8 on other cell types or in test tubes, it is conceivable that BPA-C8 may act directly on actin filaments owing to its chemical nature as a multi-cationic compound that can bind and thereby link basic residues on actin filaments [61,62,63,64,87]. Alternatively, BPA-C8 can indirectly affect the actin dynamics by shifting the metabolism of phosphatidylinositol and its derivatives such as PI(4,5)P2, which plays the central role in modulating the activity of actin-binding proteins and thereby influences the actin cytoskeleton [64,87].

All of these observations suggest that BPA-C8 does not penetrate the egg membranes to diffuse freely into the cytoplasm of sea urchin eggs.

Even if BPA-C8 does not diffuse into the egg cytoplasm, the binding onto the egg surface may transmit its effect on molecular events in the subplasmalemmal region. Firstly, it is noted that the fluorescence signals representing the subplasmalemmal actin filaments visualized by LifeAct-GFP are much more reduced in the denuded eggs compared to the intact eggs [Figure 5A, time 0]. Thus, the removal of the JC and VL tightly apposed to the egg plasma membrane affects microvillar morphology (Figure 4) and the structure on the other side of the egg membrane: the subplasmalemmal actin cytoskeleton (Figure 5A). Secondly, the CW in the fertilized eggs was lowered in amplitude if the JC and VL were removed. In intact eggs, preincubation with 1 mM BPA-C8 before insemination also significantly repressed the amplitude of the Ca^2+^ wave at fertilization. However, this effect was nullified when the same experiment was performed with denuded eggs (Figure 8 and Table 3).

Furthermore, without the JC and VL, the denuded eggs tended to be polyspermic at fertilization, as shown previously [39], but exposing the eggs to 100 μM BPA-C8 for 20 min before insemination resulted in a reduced polyspermy rate (Figure 7 and Table 2). It is conceivable that, by restructuring the surface of the denuded eggs (Figure 3), BPA-C8 somehow restored the nature of the egg surface to the state of the intact eggs by limiting the number of sperm interaction sites on the egg plasma membrane. While it is estimated that the surface of a sea urchin egg may have over a million receptor molecules for sperm binding [97], our observation implies that the JC and VL may modulate the fusion of the fertilizing sperm with the egg plasma membrane by masking the sperm-binding sites [39]. Since the ability of BPA-C8 to reduce the number of sperm entries in denuded eggs is more evident when the polyamine was added to the media than when it was microinjected (Figure 7 and Table 2), it is likely that the targets of BPA-C8 in effecting the monospermic fertilization of denuded eggs reside on the egg surface. Hence, some unknown sperm receptors distinct from 350 kDa, which is susceptible to DTT-mediated reduction [34], appear to work on the egg surface so that the presence of the JC and VL inhibits its sperm binding.

The results above suggest that the JC and VL permit the interaction and fusion of the fertilizing sperm at a preferential site on the egg plasma membrane apart from the fine controls taken by the microvillar and subplasmalemmal actin filaments, which make a direct contribution to the sperm-induced Ca^2+^ signals and entry in echinoderm eggs [49,54,82,98]. In line with this, it has been suggested that synthetic polyamine like BPA-C8 facilitates actin polymerization in the lamellipodia of fibroblasts and *in vitro* [61,62]. Actin is an essential structural and dynamic component of mediating cell contraction, cytokinesis, migration, and so on by the use of the mechanical force generated by polymerization itself or by the action of its associated proteins, like myosin [99,100,101,102,103,104]. Since polyamines are inside the cell, it is interesting to know their relationship with the actin cytoskeleton. As aforementioned, studies *in vitro* indicated that polyamine can induce actin polymerization [61,62]. Whether or not similar things happen in a living cell is less known, but it was demonstrated that exogenous spermine microinjected into *Xenopus* eggs could induce cytokinesis acting on the cortical microfilaments without fertilization or egg activation [105]. Whereas BPA-C8, a synthetic polyamine used in this study, enhanced the expansion of lamellipodia in fibroblasts [64], our results indicated that BPA-C8 generally tended to reduce actin polymerization at high doses and interfered with the polymerization dynamics of actin in the presence of JAS (Figure 6).

Although starfish and sea urchins belong to the phylum Echinodermata, many aspects of their sexual reproduction differ. Firstly, their eggs are fertilized at different meiotic stages [50,106,107]. In starfish, the sperm head extends an impressively long acrosomal process filled with actin filaments upon contact with the components of the JC [108,109,110,111]. The tip of the long acrosomal process, which measures the thickness of the JC by passing through the thick meshwork of the JC and a pore in the VL, releases Ca^2+^ in the egg upon fusing with the plasma membrane [53,54]. By contrast, the acrosomal process on the sea urchin sperm is much shorter than the length of its cone-shaped head when viewed in most EM and fluorescence microscopic images captured in the morphological studies on diverse sea urchin species [18,26,57,111,112]. Owing to these morphological differences, the formation of the acrosomal process in sperm *in vitro* and *in vivo* has been more easily documented in starfish [53,113,114,115]. In contrast, the exocytotic exposure of bindin was demonstrated mainly in sea urchin sperm by antibodies against it [16,116].

In this study, we showed that *P. lividus* sperm did not have much problem fertilizing intact eggs with a JC altered by BPA-C8 and denuded eggs after removing the JC, which was supposed to be essential for chemotaxis and acrosomal reaction. In denuded eggs, the VL was also removed; thus, the conventional sperm receptors should be disabled altogether. The conspecific sperm had no difficulty binding to the egg plasma membrane and triggering the early physiological Ca^2+^ changes, albeit altered ([39] and this contribution). Indeed, multiple sperm fused and entered the egg, as shown in this study. Hence, our observations suggest that other physical contact points on the egg plasma membrane might be distinct from the conventional sperm receptors that are reportedly susceptible to reducing agents and to the destruction of the VL. The contact between the fertilizing sea urchin sperm and the egg plasma membrane may be allowed at a preferential site by the passage of the tip of the sperm through a pore in the VL, as occurs in starfish [54]. In this regard, it is worth noting that for many animal groups, successful sperm–egg interaction and fusion are ensured by the organization of the egg and its investments to which the sperm have been adapted [4,117,118,119]. Our results may indicate the possibility of a jelly canal (channel), as first reported by Boveri in 1901 [120] in freshly ovulated *P. lividus* eggs (the same sea urchin species used for our experiments) through which the fertilizing sperm travel to reach and fuse with an egg plasma membrane site not covered with the VL, even if the visualization of the lifting of the VL far from the identified jelly canal subsequently provided a view contrary to Boveri’s idea [77].

In this regard, further studies using denuded eggs treated with lectins (plant proteins that recognize and bind with a high degree of stereospecificity to various sugar structures of glycoconjugate complexes) will shed light on the location and role of the receptor glycoproteins on the egg plasma membrane involved in the species-specific recognition and binding between *P. lividus* gametes.

## 5. Concluding Remarks

Fertilization is a multistep process in which the sperm must adhere and fuse with the egg plasma membrane and trigger a fertilization response suitable for sexual reproduction. During evolution, the structure and composition of the extracellular matrix surrounding the egg have been differentiated to activate the sperm and to carry out species-specific interaction and the fusion of gametes.

In sea urchins, the currently accepted view is that the exposure of the adhesion protein bindin on the jelly-coat-induced acrosomal process following the dehiscence of the acrosome mediates specific sperm adhesion to the vitelline layer covering the egg plasma membrane. However, our results on the denuded sea urchin eggs have shown that the jelly coat (JC) and vitelline layer (VL) do not play essential roles in inducing the sperm acrosome reaction or in mediating species-specific recognition between the fertilizing sperm and the egg, as judged by the fact that denuded eggs devoid of both layers had no difficulty in interacting conspecific sperm to the extent that they were penetrated by multiple sperm. Likewise, the alteration of the structural organization of these two layers induced by the incubation of intact *P. lividus* eggs with 1 mM BPA-C8 (Figure 2C and Figure 5) does not prevent monospermic fertilization, albeit modified, adding weight to the idea that species-specific recognition resides at the plasma membrane level.

Considering the importance given in the literature to the incorporation of sperm receptors in the VL of sea urchin eggs as an essential evolutionary advancement, our findings with denuded eggs may suggest future directions for further investigation in reproductive physiology to seek common molecular mechanisms for the fertilization process of all animal species despite striking differences in the structure of the gametes.

In conclusion, it could be said that there is still much to be learned about the molecular events comprising the fertilization process in sea urchin eggs. For instance, using Raman spectroscopy, we have recently shown that the depolymerization of the cortical actin filaments can be detected in sea urchin eggs as early as 15 s after insemination [58]. These early microvillar and cytoskeletal changes at the surface of fertilized eggs correspond to what was described in the early work before actin was ever discovered: “……*momentary loss of resistance through the normal break-down of material in the ectoplasm which pushes off the vitelline membrane*…” [121] p. 111, promoting the electrophysiological changes in the egg membrane potential concurrent with the Ca^2+^ influx. In terms of the crucial underlying role of microvillar morphology in the modulation of membrane potential and Ca^2+^ increases in starfish and sea urchin eggs [54,82,107,122,123,124], future studies are necessary to unveil the molecular mechanisms by which the fertilizing *P. lividus* sperm, by interacting with the egg plasma membrane, transduce Ca^2+^ signaling and the developmental program to form a new organism.

## Figures and Tables

**Figure 1 cells-13-01477-f001:**
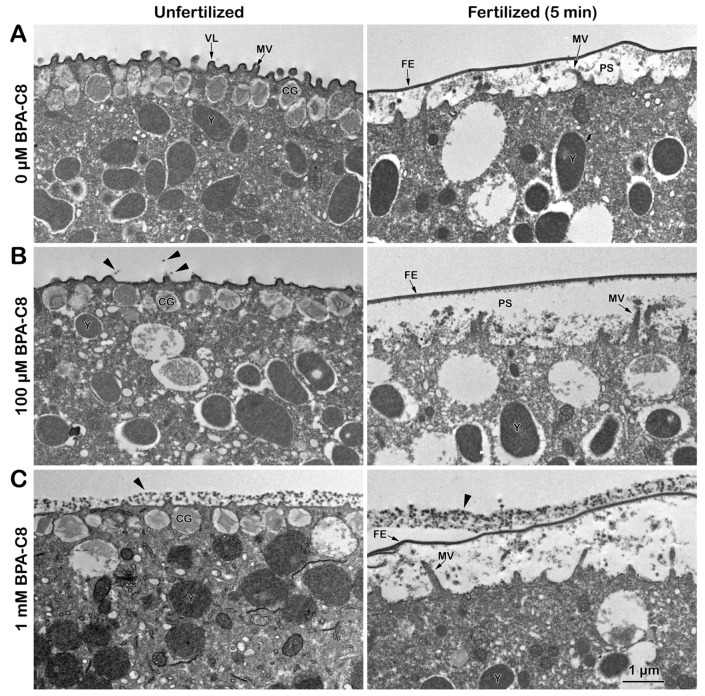
Effect of BPA-C8 on the surface of sea urchin eggs visualized with transmission electron microscopy (TEM). *P. lividus* unfertilized eggs were preincubated in NSW for 20 min in the absence (**A**) or presence of 100 µM (**B**) or 1 mM BPA-C8 (**C**). For each condition, an aliquot of eggs was fixed immediately for TEM (left column), and the remaining eggs were fertilized first and fixed in the same way 5 min after insemination (right column). Abbreviations: VL, vitelline layer; MV, microvilli; CG, cortical granules; Y, yolk granules; FE, fertilization envelope; PS, perivitelline space; arrowhead, jelly coat condensed.

**Figure 2 cells-13-01477-f002:**
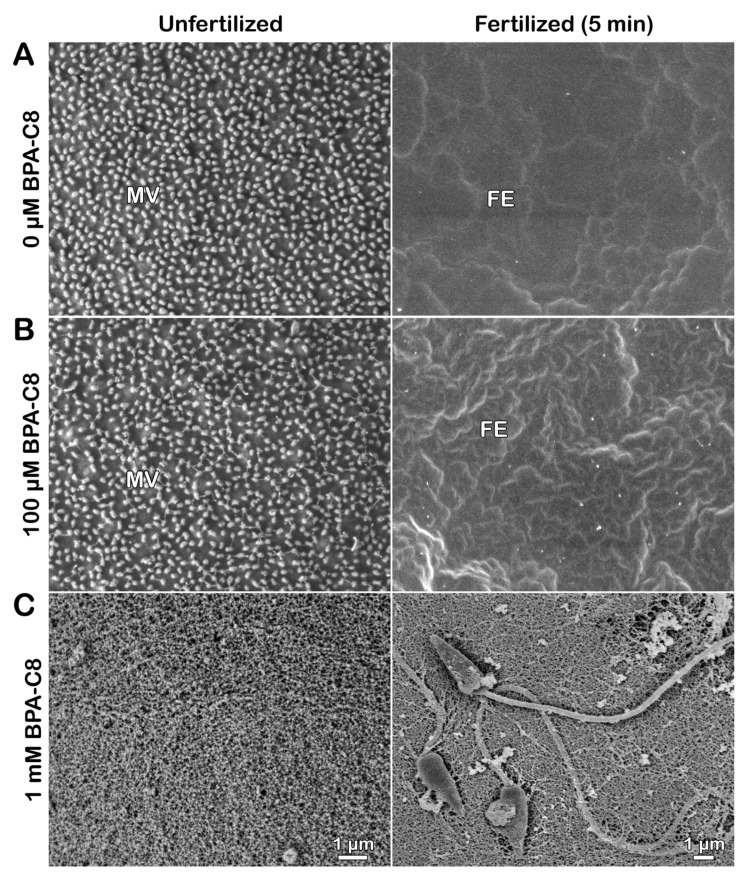
Effect of BPA-C8 on the surface of sea urchin eggs visualized with scanning electron microscopy (SEM). *P. lividus* unfertilized eggs were preincubated for 20 min in NSW in the absence (**A**) or presence of either 100 µM (**B**) or 1 mM BPA-C8 (**C**). For each condition, an aliquot of eggs was fixed immediately for SEM (left column), and the rest of the eggs were fertilized first and fixed in the same way 5 min after insemination (right column). Note the fibrous extracellular bridges forming among neighboring microvilli (MV) in B and the condensed jelly coat covering MV in C. FE, fertilization envelope.

**Figure 3 cells-13-01477-f003:**
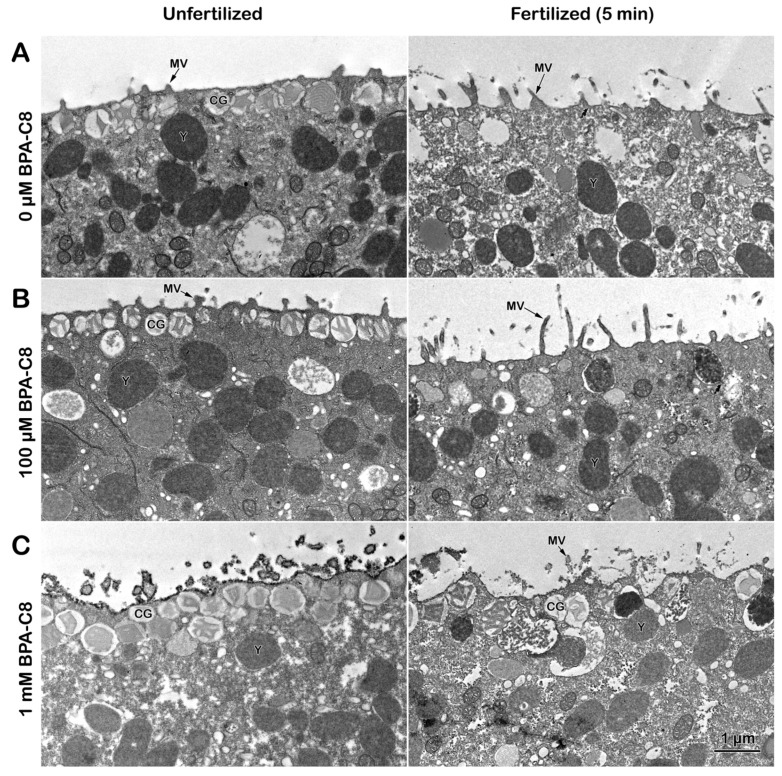
Effect of BPA-C8 on the denuded eggs of sea urchin visualized with TEM. *P. lividus* unfertilized eggs were denuded as described in the Materials and Methods section and incubated for 20 min in NSW containing 0 µM (**A**), 100 µM (**B**), or 1 mM BPA-C8 (**C**). An aliquot of eggs was fixed for TEM (left panels), and the remaining eggs were inseminated with sperm and fixed in the same method 5 min later (right panels). Abbreviations: MV, microvilli; CGs, cortical granules; Y, yolk granules. Note the absence of the fertilization envelope (FE) elevation due to removing the VL from the unfertilized eggs.

**Figure 4 cells-13-01477-f004:**
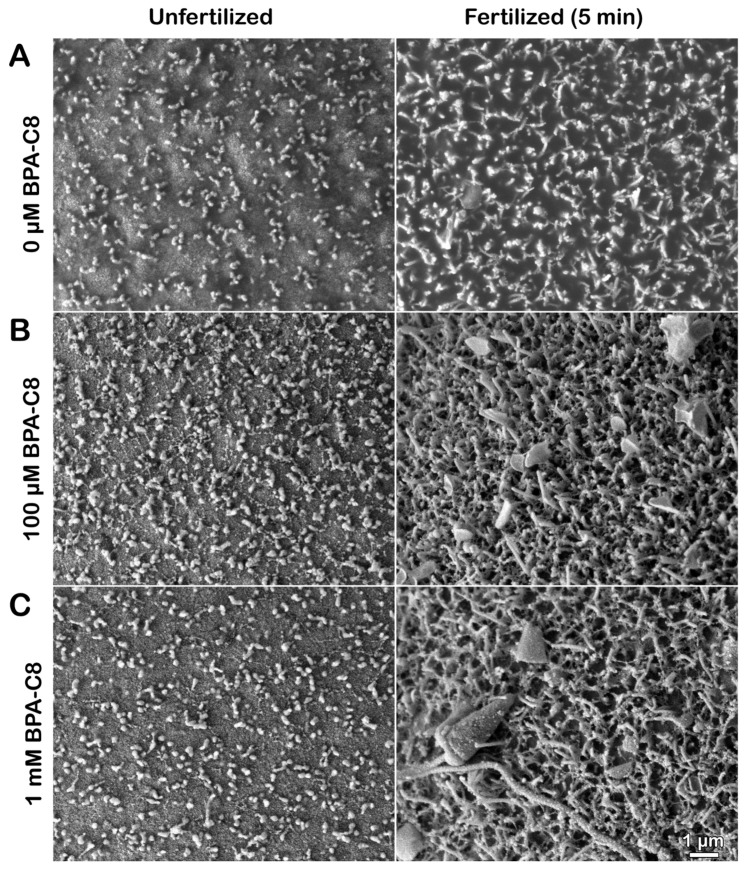
The effect of BPA-C8 on the denuded eggs of sea urchins was visualized with SEM. *P. lividus* eggs were dejellied as described in the Materials and Methods section and incubated for 20 min in NSW containing 0 µM (**A**), 100 µM (**B**), or 1 mM BPA-C8 (**C**). An aliquot of eggs was fixed for SEM (left column), and the remaining eggs were inseminated with sperm and fixed in the same method 5 min later (right column). Note the absence of the fertilization envelope (FE) elevation due to removing the VL from the unfertilized eggs.

**Figure 5 cells-13-01477-f005:**
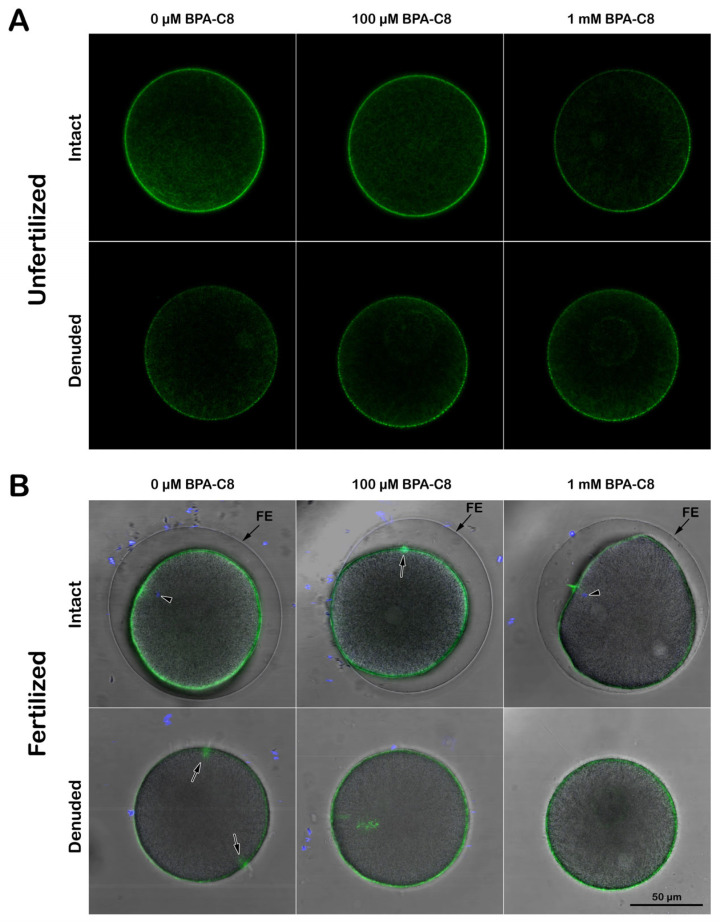
The structure of the cortical actin cytoskeleton in sea urchin eggs treated with BPA-C8. (**A**) To visualize the structural changes of the cortical filamentous actin (F-actin), *P. lividus* unfertilized intact eggs were microinjected with LifeAct-GFP (10 µg/µL in the pipette), and an aliquot of them was incubated for 20 min in NSW containing 0 µM, 100 µM, or 1 mM BPA-C8. The fluorescent images of the changes in the cortical actin cytoskeleton in the eggs induced by BPA-C8 treatment were captured by confocal microscopy at the equatorial plane (upper row). For comparison, the same batch of eggs microinjected with LifeAct-GFP was deprived of the VL and JC before incubating with BPA-C8, and images were acquired by confocal microscopy (lower row). (**B**) The changes in the cortical actin cytoskeleton in the eggs after fertilization. An aliquot of eggs treated in the same experimental condition was inseminated with sperm stained with Hoechst-33342 after dilution in NSW, and the images were captured 5 min later by confocal microscopy. The bright field view images were merged with the confocal fluorescence images to show the presence or absence of the elevated fertilization envelope (FE), the fertilization cone (arrow), and the sperm inside the egg (arrowhead).

**Figure 6 cells-13-01477-f006:**
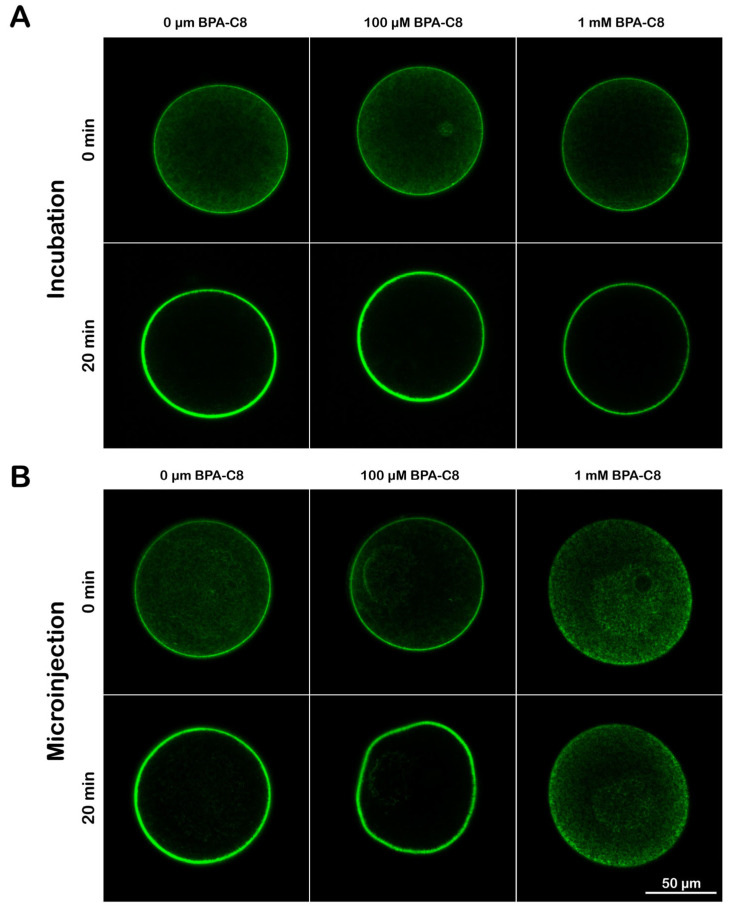
Effect of BPA-C8 on the actin dynamics inside sea urchin eggs. *P. lividus* intact unfertilized eggs were microinjected with the injection buffer containing LifeAct-GFP (10 µg/µL, pipette concentration) and then treated with varying concentrations of BPA-C8 (0, 100 µM, or 1 mM, final concentration) by bath incubation (**A**) or by microinjection (**B**). After 5 min, the eggs pretreated with BPA-C8 were incubated for 15 min with or without 12 μM jasplakinolide (JAS). The fluorescent images of LifeAct-GFP in the eggs were captured by confocal microscopy before and after the incubation with the polyamine and the actin drug (20 min total).

**Figure 7 cells-13-01477-f007:**
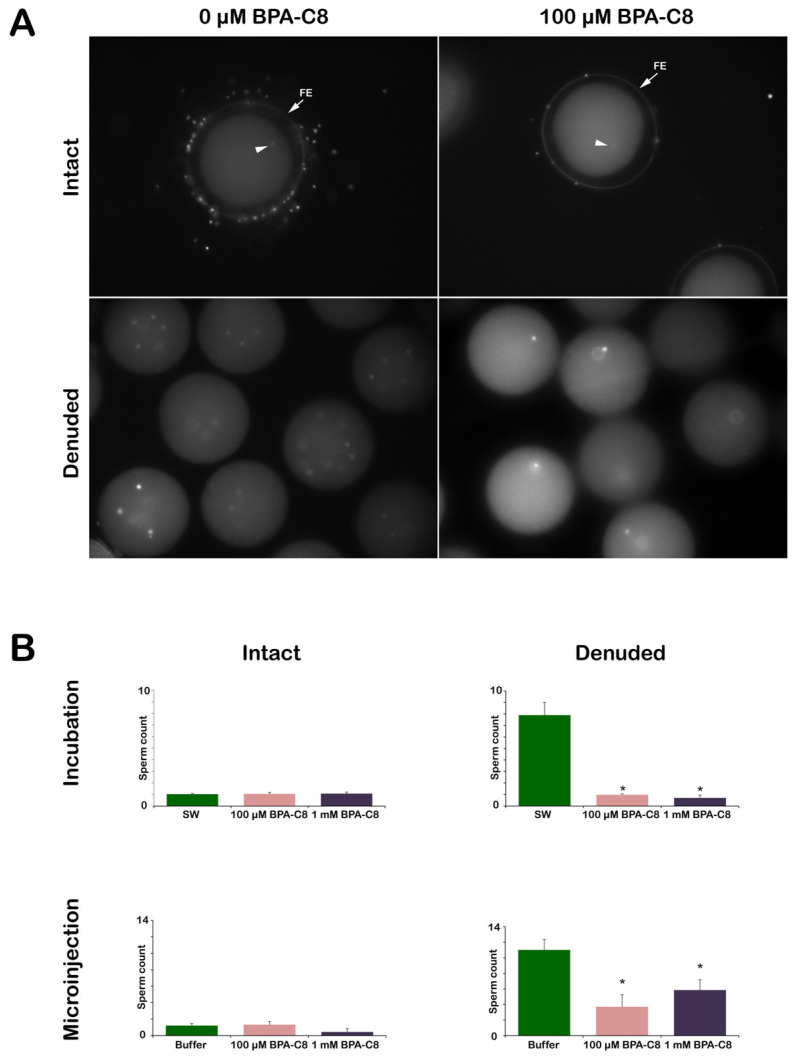
Effect of BPA-C8 on sperm entry into intact and denuded eggs. *P. lividus* unfertilized eggs were pretreated for 20 min with 0 µM, 100 µM, or 1 mM BPA-C8 either by bath incubation or microinjection. The eggs were subsequently inseminated with sperm stained with Hoechst-33342 after dilution in NSW. (**A**) Representative epifluorescence images of the internalized sperm in the intact or denuded eggs treated with BPA-C8 by bath incubation before fertilization. Sperm were counted in the eggs 5 min after insemination. (**B**) The number of sperm internalized in the intact and denuded eggs pretreated with BPA-C8 by bath incubation before fertilization. Egg-incorporated sperm were counted five minutes after insemination in the intact and denuded eggs microinjected with BPA-C8 before fertilization. The estimated final concentration of BPA-C8 in the cytosol is indicated in the histograms. * *p* < 0.01.

**Figure 8 cells-13-01477-f008:**
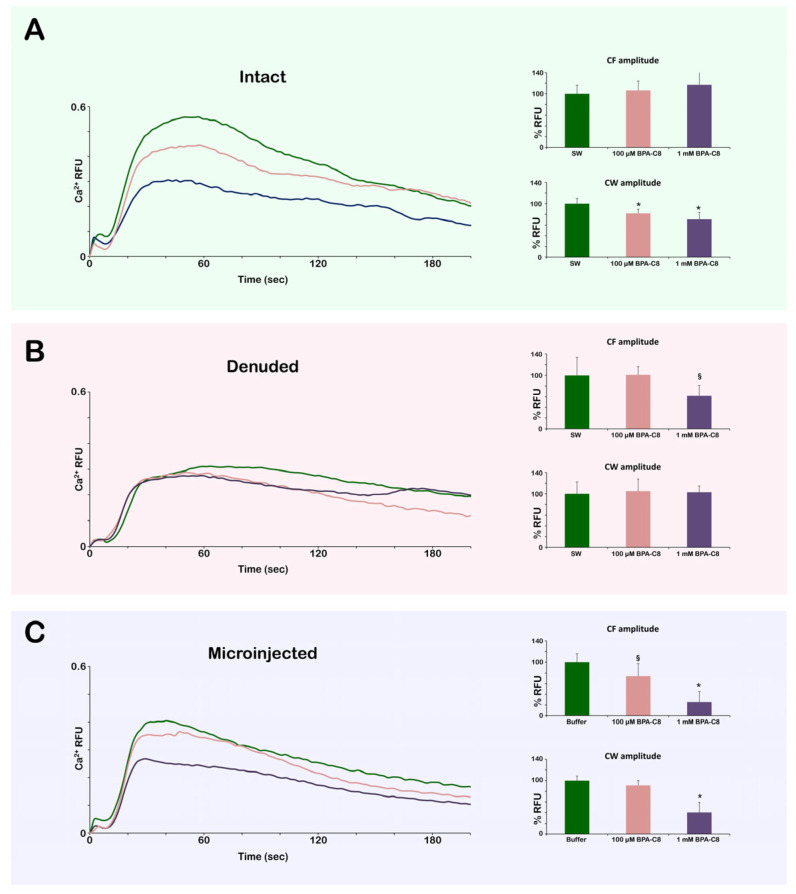
Effect of BPA-C8 on Ca^2+^ responses of the intact and denuded eggs at fertilization. Intact or denuded unfertilized eggs of *P. lividus* microinjected with Ca^2+^ dyes were pretreated with 0 µM, 100 µM, or 1 mM BPA-C8 for 20 min by bath incubation or microinjection. (**A**) Ca^2+^ trajectories in the intact eggs fertilized after the 20 min bath incubation with 0 μM, 100 μM, and 1 mM BPA-C8. (**B**) Ca^2+^ trajectories in the denuded eggs fertilized after the incubation with BPA-C8 in the same condition as with intact eggs. (**C**) Ca^2+^ trajectories in the intact eggs fertilized after microinjection with 0 μM, 100 μM, and 1 mM BPA-C8 (cytosol concentration). For Ca^2+^ trajectories, the representative results from one batch of experiments are presented, whereas the histograms are based on the numerical data pooled from all experiments with the same experimental conditions (Table 3). * *p* < 0.01, ^§^
*p* < 0.05.

**Figure 9 cells-13-01477-f009:**
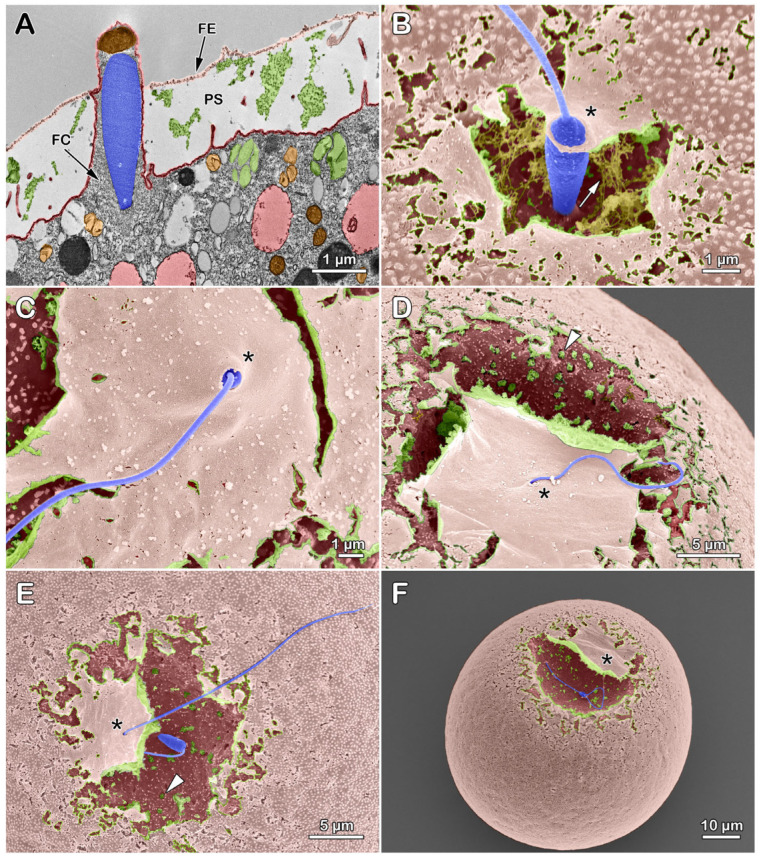
Elevation of the fertilization envelope at the early stage of fertilization in sea urchins. *Arbacia lixula* (**A**) and *P. lividus* eggs (**B**–**F**) were inseminated with conspecific sperm and subjected to a series of fixations for ultrastructural analysis for TEM (**A**) and SEM (**B**–**F**). The fixative (0.5% glutaraldehyde) was added to the samples about 10 s after insemination, and the subsequent fixation steps for TEM and SEM were followed as described in the Materials and Methods section. Abbreviations: FC, fertilization cone; FE, fertilization envelope; PS, perivitelline space. Arrowheads, the contents of the cortical granules released into the PS by exocytosis. Arrow, fibrous structures in the PS. The asterisks mark the “elastic” holes through which the sperm head and tail enter the egg.

**Table 1 cells-13-01477-t001:** Microinjected BPA-C8 interferes with jasplakinolide-induced actin polymerization.

				BPA-C8 Concentration	
*P. lividus* Eggs	BPA-C8 Treatment	JAS Incubation	0 μM	100 μM	1 mM
Intact	bath	12 μM	2.6 ± 0.21	2.4 ± 0.23	2.1 ± 0.07
Intact	microinjection	12 μM	2.9 ± 0.22	3.6 ± 0.23	0.9 ± 0.28 *

Note: Intact *P. lividus* eggs microinjected with LifeAct-GFP were treated with various doses of BPA-C8 and subsequently exposed to 12 μM jasplakinolide (JAS) for 15 min. Confocal microscopy images were captured before and after the incubation with JAS as described in the legend of Figure 6. Along the horizontal lines across the randomly positioned eggs, line scanning was performed to quantify the fluorescence intensity within the region of interest defined in the Materials and Methods section. The fold increase of the fluorescence signals (i.e., extent of enhanced actin polymerization) was calculated and reported in the table as mean ± SD based on the analyses on three different eggs for each condition. * Tukey HSD test: The average is significantly lower (*p* < 0.01) than those of 0 μM and 100 μM BPA-C8. All other pairwise comparisons were statistically insignificant.

**Table 2 cells-13-01477-t002:** Effect of BPA-C8 on sperm entry into sea urchin eggs at fertilization.

			BPA-C8 Concentration	
*P. lividus* Eggs	Treatment	0 μM	100 μM	1 mM
Intact	bath	1.02 ± 0.06 (80/80)	1.05 ± 0.13 (100/100)	1.07 ± 0.12 (80/80)
Denuded	bath	7.9 ± 1.1 (40/40)	0.97 ± 0.08 (39/40) *	0.70 ± 0.25 (28/40) *
Intact	microinjection	1.2 ± 0.25 (60/60)	1.32 ± 0.41 (60/60)	0.45 ± 0.39 (22/60)
Denuded	microinjection	11 ± 1.35 (40/40)	3.7 ± 1.55 (34/40) *	5.85 ± 1.33 (38/40) *

Note: The fluorescently labeled sperm internalized into eggs were counted 5 min after insemination by using epifluorescence microscopy. The average number of sperm per cell was expressed as mean ± standard deviation. Numbers in parentheses indicate the success rate of fertilization: number of eggs internalizing sperm/total number of eggs examined for the given condition. * *p* < 0.01.

**Table 3 cells-13-01477-t003:** Effect of BPA-C8 on the Ca^2+^ responses in the intact and denuded eggs at fertilization.

				BPA-C8 Concentration	
*P. lividus* Eggs	Treatment	Ca^2+^ Response	0 μM	100 μM	1 mM
Intact	Bath	CF	100 ± 16.1 (18)	106.5 ± 17.5 (24)	117.0 ± 28.4 (25)
Intact	Bath	CW	100 ± 10.0 (24)	81.7 ± 7.9 (28) *	70.7 ± 13.1 (27) *
Denuded	Bath	CF	100 ± 33.7 (13)	101.1 ± 14.9 (13)	61.8 ± 19.2 (12) ^§^
Denuded	Bath	CW	100 ± 22.4 (13)	105.0 ± 22.8 (13)	102.9 ± 11.6 (12)
Intact	microinjection	CF	100 ± 15.9 (15)	74.3 ± 23.0 (17) ^§^	25.6 ± 19.4 (15) *
Intact	microinjection	CW	100 ± 8.7 (15)	91.1 ± 8.9 (17)	40.3 ± 18.8 (15) *

*Note*: * *p* < 0.01, ^§^
*p* < 0.05.

## Data Availability

Data are contained within the article and Appendix A.

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
