# Peer review of "The Effect of Synthetic Polyamine BPA-C8 on the Fertilization Process of Intact and Denuded Sea Urchin Eggs"

_cells, 2024, doi:10.3390/cells13171477_

Round 1

Reviewer 1 Report

Comments and Suggestions for Authors

In this paper, in order to clarify the role of the vitrine layer and jelly coat in fertilisation, the egg surface ultrastructure, cytoskeletal actin structure, sperm penetration and calcium waves before and after fertilisation were analyzed in sea urchin normal and denuded eggs treated with synthetic polyamine BPA-C8. BPA-C8 incubation strongly suppressed polyspermy in denuded eggs and also suppressed calcium waves at fertilisation. Microinjected BPA-C8 also suppressed polyspermy of the denuded eggs and also suppressed calcium waves at fertilisation. Both bath incubation and microinjection of BPA-C8 also affected actin skeletal dynamics in eggs. The results are clear and supply important insights into the dynamic changes that occur on the egg surface during fertilisation. Some explanations appear redundant and unclear. Improvements are needed with regard to the following points.

Introduction is redundant. It should be more concise.

By what mechanism does microinjected BPA-C8 inhibit actin polymerisation and localisation on the egg surface and suppress polyspermy? What is the ultrastructure of the egg surface layer of microinjected BPA-C8?

How do authours think BPA-C8 substitutes for the respective roles of JC and VL? 

With regard to calcium waves, please explain the difference between CF and CW and the significance of each.

Please add an explanation of the relationship between Fig 9 and BPA-C8 as it was not clear.

Can Fig 5B also show only the fluorescence image? Why is only this figure superimposed on the bright field?

In the caption of Fig 6, there is mention of DMSO treatment, but is it not included in the figure?

What do the arrowheads and arrows refer to in Fig 9?

Author Response

Authors’ Responses to the Referees’ Comments

 Reviewer #1 Comments and Suggestions for Authors

We thank the reviewer for the points raised that helped improve the manuscript.

Comment 1: Introduction is redundant. It should be more concise.

Response 1: We thank the reviewer for pointing this out. We have made the introduction more concise by eliminating unnecessary redundancy.

Comment 2: By what mechanism does microinjected BPA-C8 inhibit actin polymerisation and localisation on the egg surface and suppress polyspermy? What is the ultrastructure of the egg surface layer of microinjected BPA-C8?

Response 2: The molecular mechanism by which BPA-C8 affects actin dynamics is unclear and may differ in different cell types. Its effect on the cortical actin cytoskeleton of sea urchin eggs was not so prominent as in the lamellipodia of fibroblasts. Nonetheless, 1 mM BPA-C8 interfered with hyperpolymerization of actin induced by jasplakinolide (Fig. 6). Based on the effect of natural polyamine (spermine and spermidine) and BPA-C8 on other cell types or in test tubes, it is conceivable that BPA-C8 may act directly on actin filaments owing to its chemical nature as multicationic compound that can bind and thereby link basic residues on actin filaments (ref. 61-64, 87). Alternatively, BPA-C8 can indirectly affect actin dynamics by shifting phosphatidylinositol metabolism, such as PIP2, which plays a central role in modulating actin-binding protein activity (ref. 64, 87). However, it was reported that BPAs slow down the dynamics of actin filaments without affecting the thermodynamics of actin-binding to ADF, implying that a more complex unknown mechanism is involved. Because of the limited information and the apparent differences displayed by BPA-C8 in sea urchin eggs and fibroblasts, it was not the case to speculate more about these scenarios than what was already stated in the manuscript. However, it is now stated that the mechanistic issue is not resolved. The ultrastructure of the intact sea urchin eggs microinjected with BPA-C8 is a curious question to answer. Still, we could not obtain the results for technical regions related to post-fixation. Because the breeding season was running out, while our primary interest in this study was about the structure and behavior of the denuded eggs, our priority had to be given to the denuded eggs. Nonetheless, the absence of the ultrastructure data on the intact eggs microinjected with BPA-C8 does not change our conclusion on the denuded eggs in our study.

Comment 3: How do authours think BPA-C8 substitutes for the respective roles of JC and VL? 

Response 3: Our results suggest that BPA-C8 not quite substitutes, but may compensate for the respective roles of JC and VL in some cases, like reverting the polyspermic tendency of the denuded eggs back to monospermy at fertilization. This is likely due to the character of polyamine as a molecular linker that connects negatively charged residues of macromolecules. Preserving the jelly layer of sea urchin eggs and altering the egg surface by BPA-C8 does not affect the functionality of the sperm, which can still activate the egg, albeit with a compromised fertilization response. Likewise, removing both layers does not prevent fertilization; conversely, denuded eggs (deprived of the jelly and vitelline layers) favor a polyspermic one.  

Comment 4: With regard to calcium waves, please explain the difference between CF and CW and the significance of each.

Response 4: The results on the effect of BPA-C8 on Ca2+ signals in intact and denuded (lacking the extracellular matrix) eggs at fertilization (Paragraph 3.6) with respect to the synchronized initial sperm-induced Ca2+ increase at the periphery (CF) and the progressive wave sweeping the cortex (CW) reflect BPA-C8 induced structural modification of microvilli and cortical F-actin.  As per the difference between CF (promoted by a Ca2+ influx) and the CW, future studies are necessary to help decipher the nature of the Ca2+ release of the CW, i.e., if it is the result of a Ca2+ entry or if it is triggered intracellularly by Ca2+ linked second messengers. It is a very controversial topic in the literature.

Comment 5: Please add an explanation of the relationship between Fig 9 and BPA-C8 as it was not clear.

Response 5: Figure 9 was provided to illustrate and iterate the initial stage of the fertilization process. There is no relationship between Fig 9 and BPA-C8, but Fig. 9 shows for the first time the sequence of events leading to sperm penetration by fixing P. lividus eggs that were fertilized in natural physiological conditions. In the eggs fixed only a few seconds after insemination, the vitelline layer, which will form the fertilization envelope (FE), has just detached from the egg plasma membrane. Such scanning electron microscopy images are not available in the literature because of the manipulation of the eggs to prevent the elevation of this extracellular layer, which would have prevented the visualization of the sperm on the egg surface. The images also provide evidence of a preferential entry site on the egg not covered by the vitelline layer through which the fertilizing sperm fuses with the egg plasma membrane without undergoing the jelly-coat inducing acrosome and vitelline layer species-specific recognition (see the results of the fertilization polyspermic response of eggs deprived of the extracellular matrix).  

Comment 6: Can Fig 5B also show only the fluorescence image? Why is only this figure superimposed on the bright field?

Response 6: The reason why the bright field was provided only for the fertilization response of intact eggs is that in these eggs, the vitelline layer (VL), which will form the fertilization envelope (FE), detaches from the egg plasma membrane as a result of the sperm-induced exocytosis of the cortical granules which release their content in the perivitelline space shown at scanning electron microscope in Fig. 9 (arrowheads). If the extracellular matrix, i.e., jelly and vitelline layers, are removed from the eggs, the VL separation cannot occur because of its absence.

Comment 7: In the caption of Fig 6, there is mention of DMSO treatment, but is it not included in the figure?

Response 7: We thank the reviewer for pointing out our mistake. The figure shows only the effect of jasplakinolide (dissolved in DMSO) on the cortical actin dynamics with and without incubation or microinjection of BPA-C8 and not DMSO treatment. For the sake of simplicity, we have deleted “(DMSO)” from the legend of the revised version of the manuscript.

Comment 8: What do the arrowheads and arrows refer to in Fig 9?

Response 8: The arrowheads show the exocytosis of the cortical granules in the perivitelline space, as described in the Results section. Since the scanning electron microscopy images depict sperm penetration, which is not synchronous in the different eggs displayed, it was possible in Fig. 9 B to visualize the formation of filament structures following the exocytosis of the cortical granules indicated by the arrowheads in D and E. We have added these explanations in the legend of Fig. 9.

Reviewer 2 Report

Comments and Suggestions for Authors

This manuscript provides a cohesive scientific contents, from setting up the research context to interpreting the findings. The study addresses a significant gap in understanding the role of synthetic polyamines in sea urchin fertilization and offers valuable insights into the molecular mechanisms involved.

Strengths:

The paper is well-structured, with a logical flow from introduction to discussion.

It provides a thorough examination of the role of BPA-C8 in sea urchin egg fertilization, supported by robust experimental data.

The study's findings contribute to the broader understanding of sperm-egg interactions and the role of extracellular matrix components in fertilization.

Areas for Improvement:

Condense some technical details in the introduction and discussion to enhance readability.

Clearly state the study's objectives in the introduction to provide a focused direction for the reader.

Emphasize the broader implications of the findings in the discussion to highlight their significance for reproductive biology.

Author Response

Reviewer #2 Comments and Suggestions for Authors

This manuscript provides a cohesive scientific contents, from setting up the research context to interpreting the findings. The study addresses a significant gap in understanding the role of synthetic polyamines in sea urchin fertilization and offers valuable insights into the molecular mechanisms involved.

Strengths:

The paper is well-structured, with a logical flow from introduction to discussion.

It provides a thorough examination of the role of BPA-C8 in sea urchin egg fertilization, supported by robust experimental data.

The study's findings contribute to the broader understanding of sperm-egg interactions and the role of extracellular matrix components in fertilization.

Authors’ response: We are grateful that the Reviewer appreciated the signification and the merit pf our manuscript. 

Areas for Improvement:

Condense some technical details in the introduction and discussion to enhance readability.

Authors’ response: We have revised the Introduction and Discussion to eliminate the element of redundancy. 

Clearly state the study's objectives in the introduction to provide a focused direction for the reader.

Authors’ response: Although the given study has many aspects of new exploration, such as the utility of synthetic polyamine and the biological roles of CF and VL, we have enforced the mission statement in the Introduction. 

Emphasize the broader implications of the findings in the discussion to highlight their significance for reproductive biology.

As suggested by the reviewer, we wrote a new section, “Concluding Remarks,” to reemphasize the significance of the main findings and their broader implications in reproductive biology. 

Round 2

Reviewer 1 Report

Comments and Suggestions for Authors

The conclusions presented in this article remain unclear. The experimental data clearly demonstrate that BPA-C8 treatment suppresses polyspermy and Ca² wave propagation during fertilization in denuded eggs. If the authors aim to report the effects of BPA-C8 on fertilization, as stated in the title, they should discuss the underlying mechanisms by which BPA-C8 interacts with the egg surface and cytoskeleton, as well as compare and contrast its roles of the jelly coat and vitelline layer.

In the concluding remarks, the authors mention the acrosome reaction; however, the connection between the study’s results and this conclusion is not well established. The discussion should more prominently highlight the significant effects of BPA-C8 on fertilization, as well as the roles of the jelly coat (JC) and vitelline layer (VL).

Additionally, the introduction is still overly lengthy and contains redundant information. The number of references cited is excessive for a research article. The authors should edit the introduction, focusing on background information directly relevant to the study.

Author Response

Comment 1: The conclusions presented in this article remain unclear. The experimental data clearly demonstrate that BPA-C8 treatment suppresses polyspermy and Ca²⁺ wave propagation during fertilization in denuded eggs. If the authors aim to report the effects of BPA-C8 on fertilization, as stated in the title, they should discuss the underlying mechanisms by which BPA-C8 interacts with the egg surface and cytoskeleton, as well as compare and contrast its roles of the jelly coat and vitelline layer.

Response 1: In response to the Reviewers’ previous comment on a similar issue, we revised the manuscript and stated that “the molecular mechanism by which BPA-C8 interferes with actin dynamics inside the sea urchin egg is not known,” and then provided two possible mechanistic explanations based on the previous works in the literature (i.e. direct effect on actin filaments or a pathway through shifted metabolism of PIP2). With not much being known, we believe that speculating more than that has a risk of misleading the readers. Now the Reviewer asks that we should discuss the mechanism by which BPA-C8 interacts with the egg surface. In our view, the answer was already provided in several places of the manuscript where it was stated that the cell surface is rich with negatively charged macromolecules like polysaccharides and BPA-C8 is likely to interact with them as a linker molecule. In our view, the question of how BPA-C8 and other synthetic polyamines do what they do to living cells is at the stage of collecting empirical data. Indeed, BPA-C8 was one of the multiple polyamines synthesized in the study of Ref. 64, which turned out to have a more prominent effect on the lamellipodia of fibroblasts. With the lack of structural data on the BPA-C8 complexed with macromolecules such as F-actin, it is difficult to make a definitive statement on the matter.    

Comment 2: In the concluding remarks, the authors mention the acrosome reaction; however, the connection between the study’s results and this conclusion is not well established. The discussion should more prominently highlight the significant effects of BPA-C8 on fertilization, as well as the roles of the jelly coat (JC) and vitelline layer (VL).

Response 2: In the revised manuscript, the concluding remark was added in accordance with the reviewers’ previous suggestions. Here, we opted to put our study in a bigger prospect, and that was why acrosome reaction came to be mentioned. This also made Fig. 9 encompassed in the discourse, which was one of the criticisms or suggestions made from the previous communication. In this concluding remark, we stated, “there are still many things to be learned about the molecular events comprising the fertilization process in sea urchin eggs.” Sometimes, a scientific article provides a definitive answer to a certain question, but some other times it raises more questions, opening the door to future investigation. The current study of ours happened to be the latter.

As suggested by the reviewer, we have highlighted the significant effect of BPA-C8 on fertilization as well as the role of the JC and VL by adding the following line in the concluding remarks section:    

“Likewise, the alteration of the structural organization of these two layers induced by the incubation of intact P. lividus eggs with 1 mM BPA-C8 (Fig. 2C and Fig. 5) does not prevent monospermic fertilization, albeit modified, adding weight to the idea that species-specific recognition resides at the plasma membrane level”.

Comment 3: Additionally, the introduction is still overly lengthy and contains redundant information. The number of references cited is excessive for a research article. The authors should edit the introduction, focusing on background information directly relevant to the study.

Response 3: We have tried our best to trim the number of citations to make the reference short, but it was quite difficult for us to do so because the manuscript addresses many different aspects of distinct disciplines spanning from organic chemistry to cell biology. Our manuscript with nine figures contains experimental data that could also be split into two papers but remained one. Hence, it was inevitable for the manuscript to have a lengthy Introduction and an equally long list of references. True, the shorter, the better, in general. However, sometimes even the experimental paper has many citations if it addresses various topics. In our view, our manuscript is outlandishly long in these two aspects.